# The Effectiveness of Nintedanib in Patients with Idiopathic Pulmonary Fibrosis, Familial Pulmonary Fibrosis and Progressive Fibrosing Interstitial Lung Diseases: A Real-World Study

**DOI:** 10.3390/biomedicines10081973

**Published:** 2022-08-15

**Authors:** Paolo Cameli, Valerio Alonzi, Miriana d’Alessandro, Laura Bergantini, Elena Pordon, Marco Guerrieri, Rosa Metella Refini, Piersante Sestini, Elena Bargagli

**Affiliations:** Respiratory Diseases Unit, Department of Medicine, Surgery and Neurosciences, University of Siena, 53100 Siena, Italy

**Keywords:** nintedanib, real-world effectiveness, idiopathic pulmonary fibrosis, familial pulmonary fibrosis, mortality

## Abstract

Background: Nintedanib is an oral multitarget tyrosine kinase inhibitor approved for the treatment of patients with idiopathic pulmonary fibrosis (IPF). Recent evidence demonstrated that nintedanib reduced functional disease progression also in subjects with non-IPF progressive fibrosing interstitial lung disease (PF-ILD). However, real-life data on the effectiveness of nintedanib in PF-ILD and familial pulmonary fibrosis (FPF) are lacking. Methods: this retrospective monocentric study enrolled 197 patients affected with IPF, PF-ILD and FPF treated with nintedanib at the Referral Centre of Siena from 2014 to 2021. Pulmonary functional tests and survival data were collected throughout the observation period for the evaluation of mortality and disease progression outcomes. Results: nintedanib treatment significantly reduced the FVC decline rate in IPF and PF-ILD subgroups, but not in FPF subjects. No significant differences were observed among the subgroups in terms of survival, which appeared to be influenced by gender and impaired lung function (FVC < 70% of predicted value). Concerning disease progression rate, a diagnosis of FPF is associated with more pronounced FVC decline despite nintedanib treatment. Conclusions: our research studies the effectiveness and safety of nintedanib in reducing functional disease progression of IPF and PF-ILD. FPF appeared to be less responsive to nintedanib, even though no differences were observed in terms of survival.

## 1. Introduction

Nintedanib is a multitarget tyrosine kinase inhibitor as it can competitively inhibit non-receptor and receptor tyrosine kinases: its principal mechanism of action relies on a competitive inhibition of fibroblast growth factor receptor 1, 2 and 3, platelet-derived growth factor receptor α and β, and vascular endothelial growth factor receptor 1, 2 and 3 [1]. The consequence of this wide inhibition is a down-regulation of pro-fibrotic and pro-angiogenic processes, leading to the reduction of resident fibroblasts and myofibroblasts in terms of numerosity and activity, reducing the extracellular matrix secretion and a restoration of pro-fibrotic/antifibrotic factor balance [2].

From a clinical point of view, the phase 2 double-blind, dose finding, placebo-controlled TOMORROW trial (ClinicalTrials.gov identifier: NCT00514683) and the following phase 3 trials, INPULSIS-1 and -2 (NCT01335464) demonstrated the efficacy of nintedanib in reducing disease progression, leading to the approval of the drug for clinical use in idiopathic pulmonary fibrosis (IPF) [3,4]. The subsequent open-label extension trials and large observational real-world studies substantially confirmed the effectiveness of nintedanib in reducing disease progression rate and the risk of acute exacerbation incidence in IPF patients [5,6,7,8,9,10,11].

Notably, the improvement of survival rate and progression-free survival was observed across different clinical (age, sex, respiratory functional parameters) and/or diagnostic subgroups (confident or working diagnosis) of patients affected with IPF [12,13]. However, no specific trials have been conducted for the evaluation of nintedanib efficacy in patients with familial pulmonary fibrosis (FPF), even though FPF is estimated to represent at least 10% of IPF [14].

Moreover, nintedanib is the only pharmacological therapy approved for clinical use in patients affected with progressive-fibrosing ILDs (PF-ILDs), as recently endorsed by international statements focused on the diagnosis and management of systemic sclerosis-interstitial lung disease (SSC-ILD) [15] and PF-ILD [16]. PF-ILDs includes all those non-IPF fibrotic ILDs that showed a clinical and progression rate, conventionally determined as a reduction of more than 5% of forced vital capacity (FVC) and/or of more than 10% of diffusion lung capacity for carbon monoxide (DLCO) in the previous 12 months and/or a significant worsening of respiratory symptoms, quality of life and fibrosis extension in a CT scan. Two phase 3 RCTs, SENSCIS (NCT02597933) and INBUILD (NCT02999178), explored the efficacy of nintedanib in reducing the progression of disease, expressed as FVC decline rate, in patients affected with SSC-ILD and PF-ILDs, respectively: the results demonstrated that the nintedanib arm was associated with more preserved lung function than the placebo arm after one year of treatment [17,18].

These promising findings led to the approval of nintedanib for the clinical use also in patients with PF-ILDs, significantly broadening the potential role of antifibrotic treatment in the management of ILDs; however, few data are currently available on the real-world effectiveness of this drug in this setting.

The principal aim of this research is to investigate the clinical effectiveness of nintedanib for the treatment of IPF, FPF and PF-ILDs in a real-life setting at Siena Regional Referral Centre for ILDs. Secondary outcomes include the comparison of effectiveness, safety and tolerability of nintedanib between these three subgroups of patients. 

## 2. Materials and Methods

### 2.1. Study Population and Design of the Study

All patients treated with nintedanib at the Regional Referral Centre for ILD of Siena from December 2014 to January 2021 were retrospectively enrolled into the study through the evaluation of electronical or paper medical records archived in our Centre and electronical database of Italian Medicine Agency, in which antifibrotic treatment was activated. Patients treated with nintedanib through compassionate grounds were included as well.

Diagnosis of IPF, FPF and PF-ILD was performed according to international guidelines and definitions in effect at the start of treatment [18,19,20]; all patients underwent specialistic evaluation and discussion by the ILD multidisciplinary group of Siena. FPF was diagnosed when more than two cases of idiopathic interstitial pneumonia (IPF or non-IPF) were identified in the same family. Multidisciplinary group included pulmonologists, radiologists, pathologists and, whenever needed, rheumatologists, cardiologists or occupational physicians. All the physicians involved in multidisciplinary discussions were experienced in the management of patients with ILDs. If clinical and/or radiological features could not allow a confident diagnosis and histologic sampling was contraindicated or not accepted by the patient, the multidisciplinary group provided a provisional diagnosis with high or low confidence. In case of a working high-confidence diagnosis of IPF, antifibrotic treatment was proposed to the patients: if accepted, these subjects were included in the study as well. According to upcoming clinical or radiological features and/or in the case of histological sampling by surgical biopsy or explant for lung transplant, both confident or provisional diagnoses could be re-assessed and modified. For the present study, we include in the database the definitive diagnosis and the diagnostic hypothesis with the highest confidence level made throughout the follow-up.

From January 2020, nintedanib was available for the treatment of PF-ILD through compassionate grounds: these patients were included in the study as well.

Demographic and clinical data, respiratory functional assessment, arterial blood gas analysis parameters (including pH, partial pressure of oxygen (paO_2_) and carbon dioxide (paCO_2_), fraction of inspired oxygen (FiO_2_)), radiological and, when available, histologic features were retrospectively collected and entered in the electronical database for statistical analysis.

All the available pulmonary function tests (PFTs), including DLCO assessments, performed throughout the follow-up were collected: if available, we included in the database also the PFTs of at least 1 year before starting therapy with nintedanib. To minimize the inter-observer and intra-observer variability and guarantee the best technical reproducibility and repeatability, we decided to collect only the PFTs performed at the Respiratory Diseases Unit of Siena. 

Study patients were considered lost to follow-up in case of:-Death;-Lung transplantation;-Interruption of the treatment due to any cause. Patients were excluded from the study in case of:-Inability or refusal to provide informed consent to participate in clinical studies;-Less than one month of antifibrotic treatment;-Previous antifibrotic treatment at baseline.

The main outcomes of the study were all-cause mortality and progression-free survival, defined as the time from start of nintedanib therapy to disease progression or death. Progression of disease was defined as time to decline of FVC > 10% and/or time to decline of DLCO > 15%, as previously described [21]. As secondary outcomes, safety and tolerability profile of nintedanib were evaluated through the incidence and severity of adverse events related to the treatment and the rate of temporary or definitive interruption of treatment.

The study was conducted in line with Declaration of Helsinki and was approved by local Ethic Committee.

### 2.2. Lung Function Tests 

The following lung function measurements were recorded according to ATS/ERS standards using a Jaeger Body Plethysmograph with corrections for temperature and barometric pressure: forced expiratory volume in the first second (FEV1), forced vital capacity (FVC), FEV1/FVC, total lung capacity (TLC), DLCO and capacity carbon monoxide lung transfer factor/alveolar volume (DLCO/AV) [22,23]. All parameters were expressed as percentages of predicted reference values. DLCO assessment was not performed in patients who were on oxygen therapy.

### 2.3. Statistical Analysis

Data were expressed as mean ± standard deviation, unless otherwise specified. Parametric tests (*t*-test and one-way ANOVA) were used to compare groups. Normality was assessed through the Shapiro test. Statistical analysis and graphs were performed and plotted using GraphPad Prism version 5.0 software for Windows (GraphPad Software, La Jolla, CA, USA). Unadjusted survival and disease progression outcome estimates were obtained using Kaplan–Meier curves, univariate and multivariate Cox regression analyses were performed as well to estimate the significance of different hallmarks in the study population through the survival function. Time-to decline FVC or DLCO was estimated through interpolation analysis of serial pulmonary function tests performed during the follow-up. Time-to-event endpoints were compared using a two-sided log-rank test. A *p* ≤ 0.05 was considered significant.

## 3. Results

### 3.1. Study Population

A total of 197 patients affected with ILD and treated with nintedanib were retrospectively recruited. The study population was composed of 150 subjects with IPF, 17 FPF and 30 PF-ILD, including eight idiopathic non specific interstitial pneumonia (NSIP), seven fibrotic hypersensitivity pneumonitis (fHP), nine connective tissue disease-ILD (CTD-ILD) (five SSC-ILD, three rheumatoid arthritis-ILD and one Sjogren syndrome-ILD), four undifferentiated ILD, one occupational exposure-related ILD (silicosis) and one fibrotic sarcoidosis. In the PF-ILD subgroup, a revision of the initial provisional diagnosis of IPF was performed in 10/30 patients, while the remaining were treated with nintedanib per compassionate grounds. Concerning the CTD-ILD subgroup, all patients were treated with nintedanib and standard pharmacological management of CTD, including immunosuppressive drugs, according to a multidisciplinary case-by-case evaluation conducted with experienced rheumatologists. Overall, among PF-ILDs, previous chronic oral steroids use was reported in six NSIP, five fHP patients and one sarcoidosis patient (80%, 71.4% and 100%, respectively).

Overall, at baseline, all patients treated with nintedanib showed, on average, a moderate impairment of lung volumes and lung diffusion capacity, expressed as FVC% and DLCO% predicted value (Table 1). Among the three subgroups, IPF patients showed a male prevalence, as opposed to a substantial equality in the other two subgroups and were older than FPF and PF-ILD, even though the statistical significance was reached only with the latter subgroup. No differences were observed in terms of smoking status among the subgroups. Baseline PFTs parameters were substantially similar among the three subgroups.

Pre-treatment PFTs were available in 37 patients (23, nine and five belonging to IPF, PF-ILD and FPF subgroup, respectively): the comparison of pretreatment FVC and DLCO reduction rate among the subgroups did not show any significant difference (Table 1).

Concerning radiological features at chest CT scan, IPF and FPF patients showed a clear predominance of a typical usual interstitial pneumonia (UIP) pattern, while an indeterminate for UIP or alternative to UIP pattern was observed in the majority of PF-ILD subjects. 

During the follow-up, FVC and DLCO measurements were available for 88, 49, 24 and 10 patients after 12, 24, 36 and 48 months of treatment (t1, t2, t3 and t4), respectively; if compared with pre-treatment values, FVC decline rate was significantly lower at t1 (*p* = 0.0374) and showed a substantial stability throughout the subsequent follow-up steps (Appendix A).

### 3.2. Subgroup Analysis: FPF

The study population included 17 patients affected with FPF (11 males, 70.3 ± 7.7 years old): among these, 10 patients showed a radiological pattern of typical UIP and were diagnosed as familial IPF, while the remainders showed a CT pattern of NSIP or indeterminate for UIP (five and two patients, respectively). In comparison with sporadic IPF, FPF patients were nearly significantly younger (*p* = 0.0534) and showed a higher percentage of females (*p* = 0.0119). 

Concerning functional disease progression, antifibrotic treatment did not appear to influence neither FVC nor DLCO decline rate, and that remained substantially unchanged with respect to the pretreatment period (*p* = 0.9104 and *p* = 0.3587, respectively) 

### 3.3. Subgroup Analysis: PF-ILD

Regarding demographic features, PF-ILD patients were on average younger and showed a higher percentage of female patients than IPF and FPF subjects, even though statistical significance was reached only in comparison with IPF subgroup. As expected, only a minority (7/30 patients) showed a radiological pattern of definite or probable UIP at CT scan. 

Nintedanib treatment was associated to a significant reduction of FVC deterioration rate with respect to pretreatment data (*p* = 0.0423), while no differences were observed in DLCO decline rate (*p* = 0.8823).

### 3.4. Outcome Analysis

At 1st February of 2022, median of survival and progression-free survival in the entire population was 999 and 683 days, respectively. During the follow-up, 80 patients died (40.6% of the entire study population; 63, nine and eight among IPF, FPF and PF-ILD subgroup, respectively) and five underwent lung transplantation (2.5%, three IPF and two PF-ILD subjects) while 10 interrupted antifibrotic treatment due to severe or incoercible side effects (5%, nine IPF and one PF-ILD). Permanent or temporary daily dose reduction was necessary in 53 patients (26.9%, of which 40 were IPF, 10 PF-ILD and three FPF subjects), due to persistent diarrhea or liver toxicity (45 and eight patients, respectively). Fatal or near-fatal adverse events were not observed in our cohort. We did not observe any differences in terms of frequency and severity of side effects among the three subgroups.

Survival analysis did not show significant differences in terms of all-cause mortality among IPF, FPF and PF-ILD subgroups (log rank test: 1.180, *p* = 0.554) (Figure 1).

Univariate Cox analysis revealed that male sex, baseline FVC < 70% of predicted value, baseline DLCO < 50% of predicted value, baseline paO_2_/FiO_2_ < 3, desaturation during the 6-min walking test (6MWT) and a radiological UIP pattern were significantly associated with a worse outcome; multivariate Cox analysis confirmed only male gender and FVC < 70% as factors significantly affecting the survival of study population (Figure 2) (Table 2).

FPF patients showed a significantly worse progression-free survival than IPF and PF-ILD subjects (log rank test: 11.77, *p* = 0.0028) (Figure 3).

Univariate Cox analysis confirmed that FPF diagnosis was associated to a significantly worse outcome, as well as male sex and baseline FVC < 70% of the predicted value. Multivariate analysis showed that progression-free survival was significantly influenced only by a diagnosis of FPF (Table 3 and Figure 4).

## 4. Discussion

This study embraces almost seven years of antifibrotic treatment use and includes a relevant number of patients affected by non-IPF PF-ILD with a small cohort of FPF patients. The sample size and the longtime of observation allowed us to select mortality and progression-free survival as the main outcomes of the study. The main findings of our study substantially confirmed the effectiveness of nintedanib in reducing the progression rate of fibrotic lung diseases, improving the life expectancy in patients with IPF and PF-ILD, as already reported by RCTs and real-life observational studies [3,13,18,24]. However, these results were not confirmed in the FPF subgroup, in which nintedanib appeared not to be effective in reducing FVC decline rate and progression-free survival. 

The positive impact of antifibrotic treatment in the management of IPF is widely accepted, since the reduction of progression rate and mortality has been repeatedly demonstrated, with no remarkable differences between pirfenidone and nintedanib [24,25,26,27]. The present study further confirms these findings, providing real-world evidence of nintedanib effectiveness in a population of considerable size and with a long observation period and data collection, which allowed us to evaluate the sustained effectiveness of the drug in the reduction of FVC decline rate. On this regard, our results are in line with other previous reports coming from real-life observational studies [5,24]; notably, the mean decline rate of FVC after the first year of treatment was also similar to those reported in open-label extension studies of the INPULSIS and TOMORROW trials [28,29,30], further underlining the effectiveness of this drug in the long-term management of IPF patients. Moreover, standing that our IPF subgroup could be considered more “fragile” than RCT populations (in terms of age, comorbidities and respiratory functional impairment), our findings are surely interesting since they support the efficacy of nintedanib in reducing disease progression also in these patients. Our results are also in line with other real-life studies conducted on this field, all of them showing demographic features of study populations comparable to ours [31,32].

Concerning PF-ILD and FPF subgroups, our study showed that these patients are generally younger and more frequently female than IPF subjects, as already reported in the literature [33]. Interestingly, at the inclusion of the study, corresponding to the start of nintedanib therapy, we did not observe any significant differences in terms of respiratory functional impairment and, more importantly, in the FVC decline rate in the year before the treatment among the three subgroups: this finding further underlines the clinical relevance of a strict respiratory monitoring of patients with non-IPF ILD, in order to promptly detect a progressive phenotype and provide an early antifibrotic treatment. The real prevalence of PF-ILD among the different disease entities underlying ILD has not been fully established yet [34,35] and it goes beyond the scope of this study: however, our findings support the need for a better awareness for non-IPF-ILD among non-respiratory physicians (e.g., rheumatologists) for the implementation of specific clinical and respiratory functional surveillance of these patients.

In terms of effectiveness, our results showed that nintedanib treatment roughly halved the FVC decline rate in the PF-ILD subgroup if compared with pre-treatment values: interestingly, our data are in line with the INBUILD and SENSCIS trials in terms of mean annual reduction of FVC [17,18], further supporting the efficacy of nintedanib in halting disease progression of PF-ILD also in a real-world setting. Due to the very recent publication of the SENSCIS and INBUILD trials, only one study has investigated the real-world effectiveness of nintedanib in a cohort of PF-ILD so far. Tzials et al. described their clinical experience with pirfenidone and nintedanib in a group of f-HP, whose diagnosis had been rediscussed after a first provisional diagnosis of IPF: the annual FVC and DLCO decline rate reported were 4.2% and 5.7%, respectively [36]. The numerically better outcomes observed in our cohort after one year of treatment (1.4% and 2.2%, respectively) may be influenced by the small size (30 vs. 14) but also by the different composition of study populations: indeed, we included also patients affected with PF-ILDs other than f-HP (mainly CTD-ILD and iNSIP), whose disease clinical course and response to nintedanib treatment has been reported to be slightly better than f-HP [35,37].

Very few data are available on nintedanib in FPF patients: a multicenter, international retrospective study assessed the safety and effectiveness of pirfenidone and nintedanib in a cohort of patients with a diagnosis of IPF and concomitant telomere-related gene mutations, confirming an improvement of FVC rate decline similar to that reported in sporadic IPF [38].

To our knowledge, this is the first real-world study evaluating the impact of nintedanib in the progression rate and mortality in a group of FPF patients: the only evidence of the effectiveness of antifibrotic treatment on this field comes from two studies, one of which is a case report, that investigated the effectiveness of pirfenidone on this setting, showing conflicting results [39,40]. In our study, we did not observe any modification in FVC and DLCO decline rate before and after treatment. Accordingly, FPF patients showed a significantly worse progression-free survival than IPF and PF-ILD; this finding was further confirmed by the multivariate Cox analysis which showed that a diagnosis of FPF was the only significant factor associated to a worse disease course. Despite the small sample size and the lack of consensus in the definition of FPF, these findings suggest that the effectiveness of nintedanib may be impaired in FPF subjects. Our results are conflicting with those reported by Justet et al. [38]. Unfortunately, in our population, genetic analysis was not available and patients were labelled as FPF only according to familial history and consequently we were not able to compare our data with those already published; moreover, reliable anamnestic data are often not disposable in the clinical practice and, therefore, the probability of unrecognized FPF patients included in the IPF and PF-ILD subgroup is anything but remote. However, the hypothesis that a worse or absent response to antifibrotic treatment in ILD patients could be related to specific genetic mutations or polymorphisms is anything but remote. Large, multicenter cohort studies have demonstrated that genotypic assessment significantly influences the disease progression rate and prognosis of ILD patients [41,42] and, accordingly, it may also affect, in a positive or negative way, the clinical response to antifibrotic drugs; this suggestion is indirectly supported by the evidence that N-acetylcysteine, a non-approved drug for IPF, showed beneficial effects in a specific subcohort of patients characterized by a specific single nucleotide polymorphism within the TOLLIP gene [43]. Therefore, in an optic of personalized medicine, our findings support the need for genetic assessment in FPF patients for a more accurate prediction and evaluation of response to antifibrotic treatment.

## 5. Conclusions

In conclusion, our research study describes and comprehensively analyzes the long experience of our Referral Centre with nintedanib treatment in the management of IPF, FPF and, more recently, of PF-ILD. Nintedanib confirmed its potential in reducing disease progression in these patients, while some concerns have raised for the management of FPF, in which nintedanib seems to be ineffective.

Larger cohorts of FPF patients need to be evaluated. Regarding PF-ILD, our real-life preliminary data are surely promising and will probably contribute to “lead the way” to the antifibrotic treatment also in this field. Waiting for the new oncoming antifibrotic drugs, nintedanib remained one of the milestones of pharmacological treatment of diffuse fibrosing ILDs and, therefore, our results provide further and intriguing insights in terms of long-term efficacy and personalization of therapy.

## Figures and Tables

**Figure 1 biomedicines-10-01973-f001:**
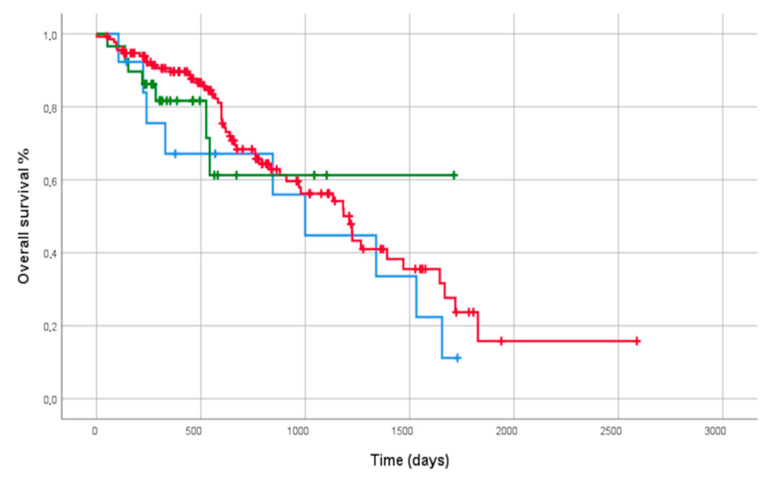
Comparison of overall survival curves of patients with IPF (**red line**), PF-ILD (**green line**) and FPF (**blue line**) through Kaplan–Meier curves.

**Figure 2 biomedicines-10-01973-f002:**
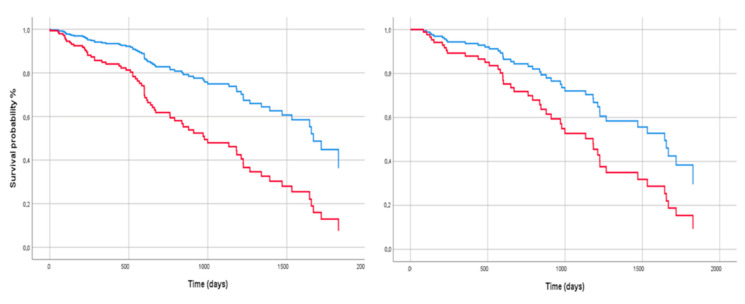
Cox regression analysis for survival of study population, stratified according to gender (on the **left**; Male: red line) and basal FVC% of predicted value (on the **right**; FVC < 70% of predicted value: red line). The covariates are reported in Table 2.

**Figure 3 biomedicines-10-01973-f003:**
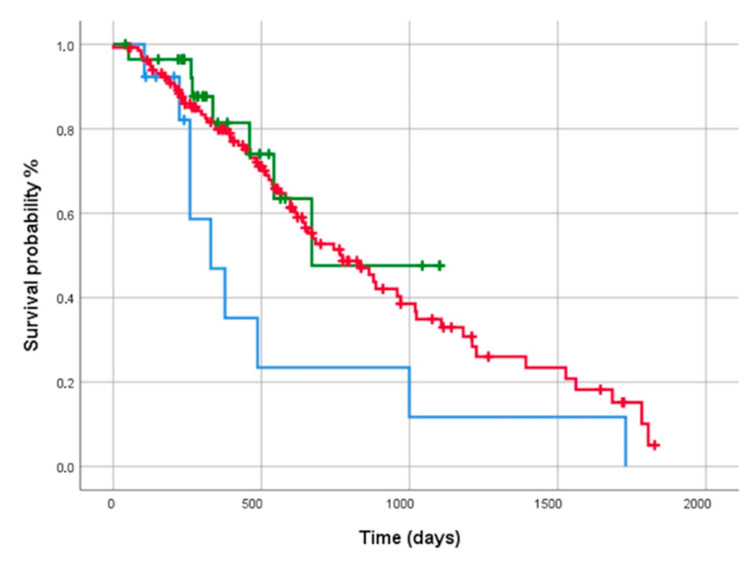
Comparison of progression-free survival among subjects with IPF (**red line**), FPF (**blue line**) and PF-ILD (**green line**), expressed with Kaplan–Meier curves.

**Figure 4 biomedicines-10-01973-f004:**
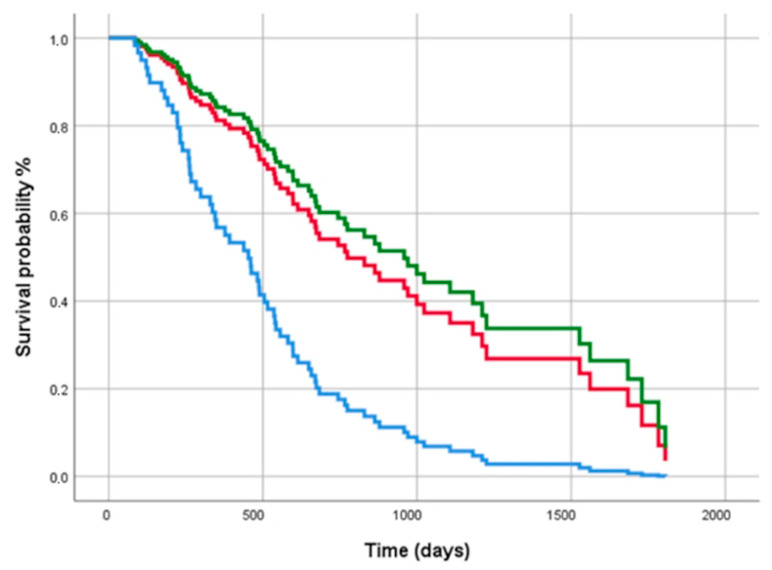
Cox regression analysis for progression free-survival among IPF (**red line**), FPF (**blue line**) and PF-ILD (**green line**) subgroups. The covariates were age, gender, basal FVC < 70% of predicted value and diagnostic subgroups.

**Table 1 biomedicines-10-01973-t001:** Demographic features, clinical data, respiratory functional parameters and radiological pattern of the entire study population and among the three subgroups. * Post test comparison between PF-ILD and IPF; ° post test comparison between FPF and IPF patients.

Parameters	Study Population	IPF	PF-ILD	FPF	*p*-Value (Mean Difference; CI 95%)
N°	197	150	30	17	*p* = 0.0019 * (6.305; 2.152–10.46)
Age (yrs)	72.6 ± 8.1	74.4 ± 8.2	68.1 ± 9.6	70.3 ± 7.7	*p* = 0.1579 ° (4.126; −2.117–11.22)
Male gender (%)	145	117 (78)	17 (56.6)	11 (64.7)	*p* = 0.0092 * (Chi-square: 9.372)
					*p* = 0.0957 ° (Chi-square: 9.372)
Pack/year	28.3 ± 15	27.2 ± 14.7	30.6 ± 17.3	25.1 ±17.5	*p* = 0.1869
*Comorbidities*					
Hypertension	81	66	10	5	*p* = 0.3283
GERD	33	25	5	3	*p* = 0.9947
Ischemic heart disease	45	33	9	3	*p* = 0.5508
Cancer	14	10	3	1	*p* = 0.8874
Diabetes mellitus	33	22	7	4	*p* = 0.3754
Osteoporosis	35	25	6	4	*p* = 0.7360
*HRCT pattern*					
UIP	121	105	6	10	*p* = 0.00004 (Chi-square: 44.77)
Probable UIP	41	40	1	0	
Indeterminate for UIP	14	5	4	5	
Not UIP	21	0	19	2	
*Baseline PFTs*					
FVC (l)	2.3 ± 0.7	2.4 ± 0.7	2.2 ± 0.7	2.1 ± 0.9	*p* = 0.0657
FVC (%)	74.2 ± 19.7	74.9 ± 18.9	71.7 ± 20.6	73 ± 22.1	*p* = 0.2817
FEV1 (l)	1.8 ± 0.5	1.9 ± 0.5	1.8 ± 0.5	1.7 ± 0.6	*p* = 0.0867
FEV1 (%)	77.5 ± 20	78.6 ± 19	75.6 ± 20.8	75.3 ± 24.1	*p* = 0.3258
TLC (l)	4.3 ± 1.1	4.4 ± 1.2	4.1 ± 0.8	4 ± 1.1	*p* = 0.0783
TLC (%)	71.2 ± 11.1	73.6 ± 17.2	68.8 ± 12.6	69.5 ± 8.5	*p* = 0.1597
DLCO (%)	42.7 ± 17.3	41.4 ± 17.9	47.6 ± 17.5	40.5 ± 15.6	*p* = 0.1357
DLCO/AV (%)	71.2 ± 23.5	68.7 ± 22.3	72.8 ± 15.6	69.7 ± 16.5	*p* = 0.3359
ΔFVC ml pre-treatment	(−256) ± 352	(−270) ± 393	(−238) ± 320.8	(−268) ± 224.5	*p* = 0.9486
ΔFVC % pre-treatment	(−8.8) ± 11.3	(−9.1) ± 12	(−7.7) ± 9.3	(−12.5) ± 11.5	*p* = 0.6589
ΔDLCO % pre-treatment	(−7.5) ± 15.7	(−7.7) ±15.1	(−5.6) ± 18.5	(−8.5) ± 12.5	*p* = 0.8853

AV: alveolar volume; DLCO: diffusion lung capacity for carbon monoxide; FEV1: forced expiratory volume in the first second; FVC: forced vital capacity; GERD: gastroesophageal reflux disease; HRCT: high resolution computed tomography; PFT: pulmonary function test; TLC: total lung capacity; UIP: usual interstitial pneumonia.

**Table 2 biomedicines-10-01973-t002:** Univariate and multivariate analysis for mortality in the entire study population.

Covariates	B	SE	*p*-Value	HR (95% CI)
**Univariate analysis**				
Age	0.015	0.015	0.330	1.01 (0.98–1.04)
Female gender	−0.935	0.334	0.005	0.39 (0.2–0.75)
Time from onset to diagnosis	−0.002	0.004	0.653	0.99 (0.99–1.01)
Diagnosis of IPF	−0.176	0.409	0.665	0.84 (0.37–1.87)
FVC > 70%	−0.605	0.278	0.029	0.54 (0.31–0.94)
DLCO > 50%	−0.669	0.341	0.045	0.51 (0.26–0.99)
paO_2_/F_i_O_2_ > 300	−1.072	0.412	0.009	0.34 (0.15–0.76)
SpO_2_ > 90% during 6MWT	−0.862	0.430	0.045	0.42 (0.18–0.98)
UIP pattern at HRCT	0.452	0.526	0.032	1.3 (0.78–2.65)
**Multivariate analysis**				
Age	0.031	0.018	0.080	1.03 (0.99–1.06)
Female gender	−0.912	0.399	0.022	0.40 (0.18–0.87)
FVC > 70%	−0.024	0.008	0.003	0.76 (0.54–0.88)
DLCO > 50%	−0.022	0.356	0.469	0.86 (0.47–1.33)
paO_2_/FiO_2_ > 300	−0.259	0.459	0.150	0.79 (0.25–1.45)
SpO_2_ > 90% during 6MWT	−0.056	0.422	0.568	0.88 (0.55–1.45)
UIP pattern at HRCT	0.156	0.436	0.458	1.2 (0.75–1.78)

6MWT: 6-min walking test; DLCO: diffusion lung capacity for carbon monoxide; FiO_2_: fraction of inspired oxygen; FVC: forced vital capacity; HRCT: high resolution computed tomography; IPF: idiopathic pulmonary fibrosis; paO_2_: arterial partial pressure of oxygen; SpO_2_: peripheral oxygen saturation; UIP: usual interstitial pneumonia.

**Table 3 biomedicines-10-01973-t003:** Univariate and multivariate analysis for progression-free survival in the entire study population.

Covariates	B	SE	*p*-Value	HR (95% CI)
**Univariate analysis**				
Age	−0.006	0.014	0.651	0.99 (0.96–1.02)
Female gender	−0.500	0.257	0.025	0.60 (0.36–1.01)
Time from onset to diagnosis	−0.005	0.004	0.225	0.99 (0.98–1.01)
Diagnosis of FPF	0.990	0.310	0.005	2.69 (1.2–6.3)
FVC > 70%	−0.392	0.242	0.030	0.67 (0.32–0.97)
DLCO > 50%	−0.159	0.279	0.568	0.85 (0.49–1.47)
paO_2_/FiO_2_ > 300	−0.719	0.455	0.114	0.48 (0.20–1.18)
SpO_2_ > 90% during 6MWT	−0.178	0.364	0.625	0.83 (0.41–1.70)
UIP pattern at HRCT	0.115	0.195	0.277	1.2 (0.72–1.65)
**Multivariate analysis**				
Age	−0.005	0.017	0.778	0.99 (0.96–1.02)
Female gender	−0.494	0.312	0.113	0.61 (0.33–1.12)
FVC > 70%	−0.361	0.266	0.175	0.63 (0.34–1.34)
Diagnosis of FPF	1.234	0.588	0.036	3.43 (1.08–10.87)

6MWT: 6-min walking test; DLCO: diffusion lung capacity for carbon monoxide; FiO_2_: fraction of inspired oxygen; FVC: forced vital capacity; HRCT: high resolution computed tomography; FPF: familial pulmonary fibrosis; paO_2_: arterial partial pressure of oxygen; SpO_2_: peripheral oxygen saturation; UIP: usual interstitial pneumonia.

## Data Availability

Not applicable.

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
