# Peer review of "The Effectiveness of Nintedanib in Patients with Idiopathic Pulmonary Fibrosis, Familial Pulmonary Fibrosis and Progressive Fibrosing Interstitial Lung Diseases: A Real-World Study"

_biomedicines, 2022, doi:10.3390/biomedicines10081973_

Round 1

Reviewer 1 Report

The article entitled"The effectiveness of nintedanib in patients with idiopathic pulmonary fibrosis (IPF), familiar pulmonary (FPF) and progressive fibrosing interstitial lung diseases (PF-ILD): a real-word study" reports a a retrospective mono-centric study on the beneficial effects of nintedanib treatment in reducing functional disease progression of IPF, FPF and PF-ILD is interesting, but many critical points are presents and it needs a major revision  

In the text there are to many abbreviations and acronym; it is  correct to define  the abbreviation and the acronym the first time that  it i9s reported in the text.

It is not reported if the patients were treated with other drugs for the comorbidities and if these treatments could affect in positive or negative the effectiveness of nintedanib.

Table 1, 2 and 3 are not clear, a more detailed legend should be reported with the explanation of abbreviations.

A plasmatic markers of fibrosis, such as fibroblast growth factor (FGF) or vascular endothelial growth factor (VEGF) should be evaluated to further investigate the pharmacological action of  nintedanib, a multi-target tyrosine kinase inhibitor.

Author Response

We would like to sincerely thank the Reviewer for providing precious comments and suggestions that will help us to improve our paper to make it suitable for publication.

Here is our point by point reply:

In the text there are to many abbreviations and acronym; it is  correct to define  the abbreviation and the acronym the first time that  it i9s reported in the text.

1- Thank you for your suggestion. We have included all the definition of abbreviations and acronyms the first time that were reported in the manuscript. Concerning the acronys of randomized clinical trials, we added also the specific clinical trials identification number.

It is not reported if the patients were treated with other drugs for the comorbidities and if these treatments could affect in positive or negative the effectiveness of nintedanib.

2- All patients were treated for medical comorbidities according to clinical indications and specialistic advices: no patients have been specifically treated for pulmonary arterial hypertension or with other drugs that have been proved to worsen IPF, PF-ILD and FPF disease progression or mortality. In CTD-ILD subjects, patients were treated with nintedanib associated to standard management of CTD, through a multidisciplinary case by case evaluation conducted with experienced rheumatologists. Oral steroids have been taken by many PF-ILD patients, including NSIP, fHP and sarcoidosis.  

We improved the Results section in order to clarify these aspects.

Table 1, 2 and 3 are not clear, a more detailed legend should be reported with the explanation of abbreviations.

3- Thank you for your comment. We have included in all the Tables a specific legend for the explanation of abbreviations.

A plasmatic markers of fibrosis, such as fibroblast growth factor (FGF) or vascular endothelial growth factor (VEGF) should be evaluated to further investigate the pharmacological action of  nintedanib, a multi-target tyrosine kinase inhibitor.

4- Thank you for your comment. The identification and validation of one or more biomarkers able to predict the response to antifibrotic treatme in IPF is still a hot topic in ILD setting. Nintedanib acts as a non-specific multi-targeted tyrosine linase inhibitor, leading to a reduction of myofibroblast proliferation and activity through the inhibition of VEGF, PDGF and FGF pathways. It is still not konown if nintedanib may lead to a reduction of VEGF, PDGF and FGF plasma levels, even though elevated concentrations of VEGF have been reported to be linked to a more aggressive phenotype of disease. The evaluation of these plasmatic markers is surely of interest and could provide intersting insights in the early prediction of good response in IPF: however, considering the retrospective nature of the study, this aspect goes beyond the scope of the present manuscript. 

Reviewer 2 Report

This study evaluated the efficacy of nintedanib in patients with IPF, familial-PF (FPF) and PF-ILD recruited and analyzed between 2014-2021 at a referral center in Siena. The authors used FVC, DLCO, and survival analysis as a readout of nintedanib efficacy in these patients. Nintedanib is a triple kinase inhibitor approved for the treatment of IPF patients. Based on pre-clinical studies, it is known to act as an anti-fibrotic therapy by inhibiting fibroblast activation and ECM gene expression. The real-world efficacy of this drug was very well-known in IPF patients however, the efficacy of this drug in familial IPF patients and PF-ILD in real-world settings is unclear. The take-home result of this study is that FPF patients are less responsive to nintedanib. But as the authors pointed out the less sample size of FPF and the unavailability of their genetics are study limitations. However, this study provides a basis for conducting more clinical studies on the efficacy of nintedanib in FPF patients, a less explored area.

The previous study that reported the efficacy of nintedanib specifically on telomere-related gene-associated FPF patients has conflicting results with the present study. Based on that fact, do authors think that nintedanib's efficacy depends on genetic variants responsible for disease progression in FPF patients? It is unknown what genetic variants are responsible for FPF in these patients, and it is very possible that they don’t belong to a particular genetic variant group. On that basis, do authors think that future studies should focus on one subgroup (genetic variants basis) of FPF patients when assessing these drug efficacies in real-world settings? Maybe it would be of value to discuss these facts/speculations in the article in an elaborated manner.

Author Response

We would like to sincerely thank the Reviewer for the comments/suggestions provided that will help us to improve our paper.

We completely agree with the Reviewer concerning our study limitations (that we already stated and commented in The Discussion) and the potential implication of our results. We believe that is currently impossible to provide a reliable answer to the questions raised by the Reviewer concerning the interpretation of our data, showing a less efficacy of nintedanib in FPF. Standing that FPF subgroup was based on anamnestic data, patients were labelled as FPF if they can "prove" to have at least one relative affected with ILD: this aspect may be influenced by the severity of relative's lung disease that may have facilitated the diagnosis. Consequently, it is feasible to hypothese that genetic mutations in these patient's clusters were those related to a more aggressive phenotype of disease. Therefore, our retrospective patients selection could be influenced by survival and reporting biases. However, in a optic a personalized medicine, FPF surely represents a hot topic in the ILD setting: considering that also a non-approved drug as NAC has showed potential beneficial effects in specific cluster of IPF patients identified through genotypic assessment, the hypothesis that a worse response to antifibrotic treatment may be related to specific genetic mutations is anything but remote. 

Therefore, following the Reviewer suggestion, we decided to improve our Discussion section as it follows:

Unfortunately, in our population genetic analysis was not available and patients were labelled as FPF only according to familial history and consequently we weren’t able to compare our data with those already published; moreover, reliable anamnestic data are often not disposable in the clinical practice and, therefore, the probability of unrecognized FPF patients included in IPF and PF-ILD subgroup is anything but remote. However, the hypothesis that a worse or absent response to antifibrotic treatment in ILD patients could be related to specific genetic mutations or polymorphisms is anything but remote. Large, multicenter cohort studies have demonstrated that genotypic assessment significantly influences the disease progression rate and prognosis of ILD patients [41,42] and, accordingly, it may also affect, in a positive or negative way, the clinical response to antifibrotic drugs; this suggestion is indirectly supported by the evidence that N-acetylcysteine, a not-approved drug for IPF, showed beneficial effects in a specific subcohort of patients characterized by a specific single nucleotide polymorphism within TOLLIP gene [43]. Therefore, in an optic of personalized medicine, our findings support the need for genetic assessment in FPF patients for a more accurate prediction and evaluation of response to antifibrotic treatment.   

Round 2

Reviewer 1 Report

The authors have done a good revision of the manuscript.